# Multiple Cryotherapy Attenuates Oxi-Inflammatory Response Following Skeletal Muscle Injury

**DOI:** 10.3390/ijerph17217855

**Published:** 2020-10-27

**Authors:** Agnieszka Zembron-Lacny, Barbara Morawin, Edyta Wawrzyniak-Gramacka, Jaroslaw Gramacki, Pawel Jarmuzek, Dariusz Kotlega, Ewa Ziemann

**Affiliations:** 1Department of Applied and Clinical Physiology, Collegium Medicum University of Zielona Gora, 65-417 Zielona Gora, Poland; b.morawin@cm.uz.zgora.pl (B.M.); e.gramacka@cm.uz.zgora.pl (E.W.-G.); 2Centre of Information Technologies, University of Zielona Gora, 65-417 Zielona Gora, Poland; j.gramacki@ck.uz.zgora.pl; 3Department of Nervous System Diseases, Collegium Medium University of Zielona Gora, Neurosurgery Center University Hospital in Zielona Gora, 65-417 Zielona Gora, Poland; p.jarmuzek@cm.uz.zgora.pl; 4Department of Neurology, Pomeranian Medical University Szczecin, 70-204 Szczecin, Poland; d.kotlega@wp.pl; 5Department of Neurology, District Hospital Glogow, 67-200 Glogow, Poland; 6Department of Sport Kinesiology, Poznan University of Physical Education, 61-871 Poznan, Poland; ziemann.ewa@gmail.com

**Keywords:** cytokines, growth factors, hydrogen peroxide, nitric oxide, whole body cryotherapy

## Abstract

The oxi-inflammatory response is part of the natural process mobilizing leukocytes and satellite cells that contribute to clearance and regeneration of damaged muscle tissue. In sports medicine, a number of post-injury recovery strategies, such as whole-body cryotherapy (WBC), are used to improve skeletal muscle regeneration often without scientific evidence of their benefits. The study was designed to assess the impact of WBC on circulating mediators of skeletal muscle regeneration. Twenty elite athletes were randomized to WBC group (3-min exposure to −120 °C, twice a day for 7 days) and control group. Blood samples were collected before the first WBC session and 1 day after the last cryotherapy exposure. WBC did not affect the indirect markers of muscle damage but significantly reduced the generation of reactive oxygen and nitrogen species (H_2_O_2_ and NO) as well as the concentrations of serum interleukin 1β (IL-1β) and C-reactive protein (CRP). The changes in circulating growth factors, hepatocyte growth factor (HGF), insulin-like growth factor (IGF-1), platelet-derived growth factor (PDGF^BB^), vascular endothelial growth factor (VEGF), and brain-derived neurotrophic factor (BDNF), were also reduced by WBC exposure. The study demonstrated that WBC attenuates the cascade of injury–repair–regeneration of skeletal muscles whereby it may delay skeletal muscle regeneration.

## 1. Introduction

Skeletal muscle performance might be temporarily impaired by high-intensity exercise. The attenuation in muscular strength may be transitory, it may last minutes, hours, or several days following training or competition [1]. The essential processes in the regeneration of injured skeletal muscles involve the leucocytes activation, the proliferation of satellite cells, and vascularization. Myogenesis and angiogenesis are a prerequisite for the subsequent morphological and functional healing of the injured muscle. This leads to rebuilding of the damaged myocytes and vessels, restoration of the blood flow, and restoration of the oxygen supply to the tissue [2]. Reactive oxygen and nitrogen species (RONS) play a key role as signal molecules and vasodilators in the activation of several growth factors such as fibroblast growth factor (FGF), vascular endothelial growth factor (VEGF), insulin-like growth factor (IGF-1), hepatocyte growth factor (HGF), platelet-derived growth factor (PDGF^BB^), and brain-derived neurotrophic factor (BDNF) which are extracellular signals regulating the functions of muscular, vascular, and nervous systems [2,3]. Hydrogen peroxide (H_2_O_2_) is produced by three members of the NADPH oxidase family (NOX4, DUOX1, and DUOX2) often as a consequence of the inflammatory response. Nitric oxide (NO) is produced by three isoenzymes called nitric oxide synthases (NOS), all present in skeletal muscles. While neuronal NOS (nNOS) and endothelial NOS (eNOS) are the isoforms which are expressed constitutively, inducible NOS (iNOS) is mainly expressed during inflammatory response. This response is an integral part of the muscle tissue repair after injury, mobilizing leukocytes that contribute to the clearance and regeneration of damaged tissue [4,5]. The time taken to return to cellular homeostasis and peak functional capacity after exercise-induced muscle damage may be related to the recovery of the cells directly damaged by exercise as well as the neighboring cells [4,6]. The inflammation-derived NO and H_2_O_2_ play a conflicting role in tissue repair. On the one hand, in combination with growth factors, they participate in muscle regeneration and repair. On the other hand, however, the local persistence of RONS sustained by infiltrated neutrophils may cause further injury by oxidatively damaging differentiating myoblasts and myotubes thus delaying the complete restoration to health [7,8].

Over recent years, cold exposure in the form of cold-water immersion (CWI) or whole-body cryotherapy (WBC) have been introduced into sports medicine to relieve pain and inflammatory symptoms associated with chronic pathological conditions but also to improve exercise performance and recovery [9]. A typical session of WBC involves the participant standing in a chamber filled with extremely cold gas at the temperature between −110 and −190 °C for 2–5 min [10]. The physiological benefits of WBC in athletes have been attributed to cold-induced analgesia, reduction of muscle temperature, and suppression of inflammation-derived RONS and cytokines. Studies into the effects of a cold therapy on exercise performance and recovery have reported diverse outcomes ranging from beneficial [11,12,13] through negligible [14,15,16,17] to negative ones [18,19]. Roberts et al. [19] indicated that post-exercise cold water immersion could even attenuate acute anabolic signaling and long-term adaptation of muscular system to exercise. Although repeated WBC may prove effective in reducing systemic markers of skeletal muscle damage, its effect on regenerative processes is mostly limited to cortisol and cytokines such as interleukin 6 (IL-6), serum interleukin 1β (IL-1β), tumor necrosis factor α (TNFα), and interleukin 10 (IL-10). Thus, it is difficult to confirm which aspects of the recovery process are affected by cryotherapy or cold-water immersion [10]. Available literature data according the combing exercise training and cold therapy do not present a clear position. Thus, the potential favorable effects of WBC exposure on the skeletal muscle recovery, especially when applied prior to exercise, and the underlying mechanisms of cryotherapy impact need further clarification. Therefore, this study was designed to investigate the impact of repeated WBC events on oxi-inflammatory mediators regulating the injury–repair–regeneration of skeletal muscles.

## 2. Materials and Methods

### 2.1. Subjects

Twelve elite male wrestlers, members of the national team, were included in the observation (Table 1). Each athlete underwent a thorough screening, including a full medical evaluation in the National Centre for Sports Medicine of Poland. The exclusion criteria included a serious injury/orthopedic injury, dietary supplements or medications intake, dehydration, and anemia detected at any point of the entire observation. The athletes participated in a 14 day training camp at the National Olympic Sport Centre of Poland during preparatory periods for the new competition season (endurance training 50%, directed training 24%, and special power training 26% of training load). Prior to the training camp, the athletes were randomly assigned to a control group (CON *n* = 9) and a group exposed to whole-body cryotherapy (WBC *n* = 11). Throughout the camp all athletes lived at the same accommodation and followed the same training schedule, sleeping time, and diet. Daily energy value of foods did not exceed 5200 kcal and the protein dose varied from 1.6 to 1.8 g/kg of body weight. During the camp, the wrestlers consumed an isotonic sports drink Vitargo (osmolality 317 mOsm/kg H_2_O) or plain water. All the subjects were informed of the aim of the study and signed a written consent to participate in the project. The protocol of the study was approved by the ethics committee at Medical University Poznan (N° 550/11), in accordance with the Helsinki Declaration.

### 2.2. Whole-Body Cryotherapy

The athletes were exposed to cryotherapy twice a day (at 8.00 a.m. and 6.00 p.m.) for seven consecutive days in 3 min WBC sessions at −120 °C, totaling 14 exposures (Chamber Creator, Wroclaw, Poland) at the Olympic Sports Centre (Figure 1). WBC was performed under the supervision of a doctor who had visual and auditory contact with the participants. Before the first WBC exposure, the participants were instructed to towel dry themselves of any sweat and they were provided with cotton gloves, socks, shoes, a headband, and a mask to protect their extremities. The participants were instructed to walk slowly around the chamber during each WBC session. The athletes randomized to the control conditions were instructed to sit at the room temperature (23 °C) for the same duration as the WBC treatment (3 min) according to the protocol of Broatch et al. [17].

### 2.3. Body Composition

Body mass (BM) and body composition, fat-free mass (FFM), and fat mass (FM) were estimated using In-Body720 (InBody Inc., Tokyo, Japan) calibrated prior to each test session in accordance with the manufacturer’s guidelines. Duplicate measures were taken with the participant in a standing position; the average value was used for the final analysis. The recurrence of measurement amounted to 98%. The measurements were taken between 7:00 and 8:00 a.m. on the 1st day of the training camp.

### 2.4. Blood Sampling

Blood samples were taken on the 1st day of the training camp (1 day before the first cryotherapy session), and then on the 9th day of the training camp (1 day after the last cryotherapy session) from the median cubital vein between 7.00 and 8.00 a.m. using S-Monovette-EDTA K_2_ tubes (SARSTEDT AG & Co. KG, Nümbrecht, Germany) for hematological analysis and S-Monovette tubes for other biochemical markers. Within 20 min, the samples were centrifuged at 3000 g and +8 °C for 10 min. Aliquots of serum were stored at −80 °C. All samples were analyzed in duplicate or triplicate in a single assay to avoid interassay variability. The intraassay coefficients of variation (CV) for the used kits were <7%.

### 2.5. Skeletal Muscle Damage

Serum total creatine kinase (CK) activity and myoglobin (Mb) concentration were used as the markers of sarcolemma disruption. CK was evaluated by means of the reagents and mobile spectrophotometer DP 310 Vario II (Diaglobal, Berlin, Germany) at the temperature of 20–25 °C. Mb concentration was measured by the Oxis Research kits (OXIS International, Inc, Portland, OR, USA) with the detection limit at 5 ng/mL. Percentage changes in CK and Mb levels between the initial level (S1) and post-whole body cryotherapy (S2) level were calculated as Λ(%) = ((S2−S1)/S1) × 100.

### 2.6. Oxi-Inflammatory Mediators

H_2_O_2_ and NO levels were measured by enzyme immunoassay and colorimetric methods using the Oxis Research kits (OXIS International, Inc, Portland, OR, USA). H_2_O_2_ and NO detection limits were 6.25 and 0.5 µmol/L, respectively. IL-1β and TNFα levels were determined by enzyme immunoassay methods using commercial kits R&D Systems (R&D Systems, Inc., Minneapolis, MN, USA). Detection limits for IL-1β and TNFα were 0.023 and 0.038 pg/mL, respectively. C-reactive protein (CRP) concentration was identified using commercial kit from DRG International (DRG International, Inc., Springfield Township, NJ, USA) with the detection limit at 0.001 mg/L. Serum HGF, IGF-1, muscle isoform of PDGF^BB^, VEGF, and BDNF were evaluated by R&D Systems ELISA kits. Detection limits were 40 pg/mL, 0.026 ng/mL, 15 pg/mL, 9 pg/mL and 20 pg/mL, respectively.

### 2.7. Hematological and Immunological Variables

The hematological markers (hemoglobin (HB), red blood cells (RBC), hematocrit (HTC), mean cell volume (MCV), mean corpuscular hemoglobin (MCH), mean corpuscular hemoglobin concentration (MCHC) and platelets (PLT)) and white blood cell counts (leucocytes (LEU), lymphocytes (LYM), neutrophils (NEU), monocytes (MON)) were determined by Diagnostyka (DIAGNOSTYKA LABORATORIA MEDYCZNE Co., Krakow, Poland).

### 2.8. Statistical Analysis

Statistical analyses were performed using the R software (R Foundation for Statistical Computing, Vienna, Austria) [20]. The assumptions for the use of parametric or nonparametric tests were checked using the Shapiro–Wilk and the Levene tests to evaluate the normality of the distributions and the homogeneity of variances, respectively. The significant differences in mean values between the groups (CON vs. WBC) were assessed by the unpaired version of *t*-test or the Mann–Whitney nonparametric test (if the normality was violated). The comparisons of repeated measurements (1st day vs. 9th day of training camp) were assessed by the paired version of *t*-test or the Wilcoxon nonparametric test. Additionally, eta-squared (*η*^2^) was used as a measure of effect size which is indicated as having no effect if 0 ≤ *η*^2^ < 0.05, a minimum effect if 0.05 ≤ *η*^2^ < 0.26, a moderate effect if 0.26 ≤ *η*^2^ < 0.64, and a strong effect if *η*^2^ ≥ 0.64 [21]. Pearson’s correlation coefficients were calculated to describe the relationships between circulating oxi-inflammatory mediators. Statistical significance was set at *p* < 0.05.

## 3. Results

### 3.1. Skeletal Muscle Damage

CK activity reached 2.5–3-fold increase on the 9th day in the CON group. The percentage changes of CK activity in CON reached 214 ± 151% (Figure 2). Mb concentration followed a similar pattern to CK activity i.e., it increased 3-fold in CON. The percentage changes of Mb concentration in CON amounted to 252 ± 105% (Figure 3). CK activity reached lower values while Mb concentration did not differ after 7 day WBC exposure compared to the control group. This indicates that repeated cryotherapy did not have a considerable impact on the extent of tissue damage in athletes.

### 3.2. Oxi-Inflammatory Mediators

Simultaneous changes in H_2_O_2_ and NO concentrations were observed and found to reach high values on the 9th day of the training camp. The 7 day WBC was found to significantly reduce the generation of H_2_O_2_ and NO (Table 2). Similarly, the inflammatory cytokines IL-1β and TNFα as well as hsCRP concentrations increased simultaneously on the 9th day of the training camp. Although cryotherapy was found to reduce circulating IL-1β and hsCRP in the study athletes, the level of TNFα remained unaffected. The value *η*^2^ indicated a strong influence of cryotherapy on H_2_O_2_, NO, IL-1β, and hsCRP concentrations. The changes to the concentration of circulating growth factors occurred in parallel following sports training and nearly all of them were dependent on skeletal muscle damage, except for IGF-1. CK activity highly correlated with HGF (r = 0.804, *p* < 0.001), PDGF^BB^ (r = 0.616, *p* < 0.01), BDNF (r = 0.474, *p* < 0.05), and VEGF (r = 0.483, *p* < 0.05) in the control group. Interestingly, WBC significantly reduced the levels of the growth factors which are extracellular signals stimulating regeneration of muscular, vascular, and nervous systems. The value *η*^2^ indicated a strong effect of WBC on the levels of growth factors, especially IGF-1 and PDGF^BB^. H_2_O_2_ and NO generation significantly modulated the release of growth factors into the circulation (Table 3). The strongest associations were observed for NO and HGF as well as NO and IGF-1.

### 3.3. Hematological Variables

No clear changes in hematological variables were observed except for HTC which increased following WBC group, mainly because of the changes in white blood cells count (Table 4). The numbers of NEU increased whereas LYM decreased after cryotherapy. The changes in white blood cells seem to be an interesting result, particularly when the systemic immune inflammation index needs to be determined.

## 4. Discussion

The skeletal muscle regeneration is a complex event which includes changes in generation of reactive oxygen and nitrogen species, interactions between skeletal muscle and the immune system as well as satellite cells activation [22,23]. In sports medicine, the efficiency of regenerative processes is decisive for athletes’ health and physical performance. Therefore, numerous therapies are used to modify the cascade of injury–repair–regeneration of skeletal muscles [24,25]. A disruption of the muscle structure after mechanical stress (e.g., high-intensity/strenuous physical exercise) and in the course of muscle degenerative diseases is reflected by an increase in CK and Mb levels [26,27]. In the present study, a 3-fold increase in CK and Mb was accompanied by high concentrations of H_2_O_2_, NO, IL-1β, TNFα, hsCRP as well as the analyzed growth factors. CK activity highly correlated with HGF, PDGF^BB^, BDNF, and VEGF in the control group. This confirms our previous observation that muscle injury is a necessary component to generate oxi-inflammatory response and tissue reconstruction [26]. WBC was not found to significantly affect the skeletal muscle damage in our athletes who were exposed to cryotherapy twice a day for seven consecutive days of sports training. According to Rose et al. [10], a reduction in circulating CK was proportional to the number of exposures to WBC during the recovery process. Both Hausswirth et al. [28] and Fonda et al. [29] found no significant changes in CK when protocols with either three or six exposures to WBC were applied. In the study conducted by Ziemann et al. [30], the participants exhibited a pronounced decline of circulating blood CK after ten exposures to WBC over a 5 day period. The discrepancy in results indicates the need for more precise methods of muscle damage evaluation and the standardization of WBC protocols used during the recovery process. Additionally, both the frequency and the intensity of training sessions should be taken into consideration before cold therapies are applied.

Under conditions which amplify or prolong the initial inflammatory response manifested by CRP increase, muscle damage can be considerably increased by NO and H_2_O_2_ produced in neutrophils and macrophages by iNOS and NADPH oxidase. An increase in NO and H_2_O_2_ production is viewed as being beneficial during exercise due to its effects on blood delivery, glucose uptake, contractility, etc. In the present study, NO and H_2_O_2_ were significantly elevated in the control group, in contrast to the athletes exposed to cryotherapy. Our study is the first one to have demonstrated that multiple WBC (totaling 14 sessions) reduced RONS production thereby leading to an attenuated response of the regenerative system including growth factors. Earlier, similar observations were only made in animal models by Vieira Ramos et al. [31] and Siqueira et al. [32]. The authors showed that cryotherapy reduced oxidative stress and inflammatory response without altering muscle regeneration processes and extracellular matrix remodeling. The effect of cryotherapy was related to an increase of enzymatic and nonenzymatic antioxidant systems [33]. However, the antioxidative effect of cold exposure is more evident at the temperature of −60 °C then −90 °C [34].

The pro-oxidative imbalance is necessary for athletes to reach adaptation to high training load. The H_2_O_2_ and NO molecules are involved in transcriptional control through redox modification or nitrosation of transcription factors which induce the expression of many molecules e.g., cytokines and growth factors [7]. IL-1β and TNFα are expressed in skeletal muscles for up to 5 days following muscle damage. Initially, they are involved in the degradation of the damaged tissue and then they are engaged in the subsequent muscle regeneration. Both cytokines enhance H_2_O_2_ and NO production whereby they amplify the signal transduction to cell nucleus [35,36]. The exercise-induced IL-1β and TNFα release may have a negative effect on striated muscle and can be associated with the symptoms of overtraining [37,38], however, this phenomenon has been recorded in very few athletes [39]. As was the case with the study by Mila-Kierzenkowska et al. [40], in our investigation the concentration of IL-1β was shown to increase by 30% in comparison to the level observed after cryotherapy treatment. The concentration of CRP was also found to rise by 80% with regard to the baseline value in controls but a decrease in CRP level was observed in the cold condition, which indicates that WBC treatment blunted the inflammatory response. Interestingly, TNFα did not respond to the 7 day WBC treatment, which is inconsistent with the findings reported by Ziemann et al. [41] who showed that a 5 day training protocol combined with WBC, applied in highly trained tennis players after the tournament season, induced a 60% decrease in TNFα. According to Rose et al. [10], the sensitivity of cytokine profile varies across the studies and it seems specific to both the baseline inflammatory status and to the type and control over the exercise stimulus.

Studies in human isolated muscle and myotube culture have demonstrated that NO and H_2_O_2_ are key regulators of pre- and posttranslational signaling events leading to cytokines, heat shock proteins, and growth factors synthesis [42]. The growth factors especially involved in myogenesis include HGF, IGF-1, PDGF^BB^, VEGF, and BDNF which are released from leucocytes and muscle cells within a few hours after muscle damage and then secreted from other tissues during the following few days. The timing and availability of these growth factors, as well as their receptors density on or within the myogenic satellite cells, are critical mediators in the regenerative process [43]. In our study, the changes in circulating growth factors occurred simultaneously following sports training and WBC exposure, however, most of the growth factors were found to be related to NO and H_2_O_2_ generation (Table 3). Cryotherapy was observed to reduce the levels of all analyzed growth factors, except for BDNF, with the changes being most pronounced in IGF-1 and PDGF^BB^ levels. The value *η*^2^ indicated that WBC produced the strongest effect on IGF-1 and PDGF^BB^ in comparison with other growth factors. Thus, application of cryotherapy may attenuate adaptive response to physical workload. According to Schoenfeld [44], exercise induces the synthesis of IGF-1 in myocytes and hepatocytes through the mitogen-activated protein kinases cascades that are central signaling pathways and regulate a wide variety of cellular processes, including proliferation, differentiation, apoptosis, and stress response. A significant reduction of post-cryotherapy changes was observed in HGF which had been proven to increase myogenic satellite cells migration to the site of injury and to play a prominent role in regulation of early phases of muscle regeneration. Its release from the muscle extracellular matrix is initially mediated via NO release after mechanical or injury-induced signals [45]. As was the case with HGF, VEGF was observed to increase the least significantly after 7 day cryotherapy. This growth factor improves skeletal muscle repair through modulation of angiogenesis, however, recent studies concerning therapeutic vascularization have demonstrated that the mechanism is regulated by PDGF^BB^ [46]. Therefore, we conclude that a decrease in the synthesis and secretion of HGF, IGF-1, PDGF^BB^, and VEGF can delay muscle regeneration in athletes.

The BDNF, in turn, is the circulating factor which deserves special attention. This growth factor is part of the neurotrophic family and is responsible for the viability and functioning of a variety of neuronal subtypes within the brain. In skeletal muscle, BDNF is accountable for proliferation and differentiation of satellite cells as well as the growth of the myofibers [47]. The available data show that almost 70–80% of circulating BDNF come from the brain and 25% from contracting muscles [48,49]. However, the mechanism of low temperature impact on neurotrophins has been poorly investigated. In this study, BDNF concentration did not change following cryotherapy exposure in contrast to the outcomes reported by Rymaszewska et al. [50] who observed an increase in circulating BDNF and an improvement of memory deficits in patients with mild cognitive impairments after multiple WBC. Some data also suggest that WBC improves sleep quality in athletes during high level standard competitions due to a greater pain relief and an increased parasympathetic nervous system activity during the slow-wave sleep period [51]. The aforementioned studies show that WBC has a multidirectional impact on muscular, vascular, and also nervous systems, therefore, WBC application in athletes should be considered with caution taking into account many circumstances including the sports discipline, the training load, and preparatory or competitive period.

## 5. Conclusions

Although passive WBC exposure was reported to have a positive influence on oxi-inflammatory mediators during sporting recovery, our findings based on the randomized controlled study demonstrated that cryotherapy attenuated the cascade of injury–repair–regeneration of skeletal muscles whereby it may induce an adverse effect through a delayed skeletal muscle regeneration. Therefore, if a combination of cold therapy and exercise training is to be recommended, the extent of exercise-induced muscle damage and possible disturbance of anabolic signaling should be taken into account.

## Figures and Tables

**Figure 1 ijerph-17-07855-f001:**
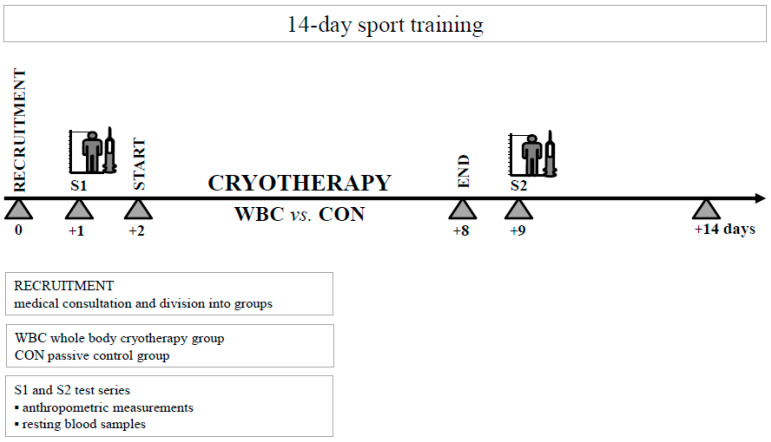
Illustration of the study design.

**Figure 2 ijerph-17-07855-f002:**
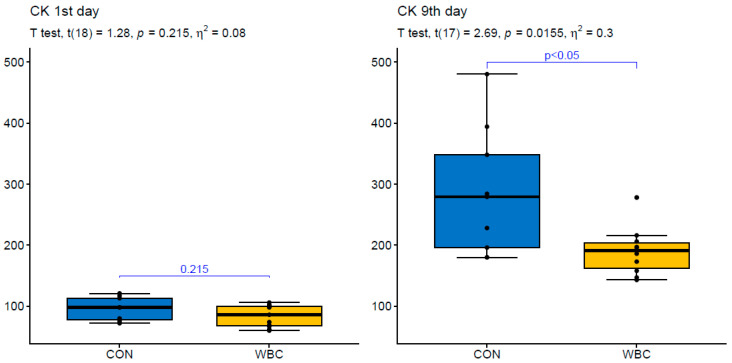
Visualization of statistical analysis of changes in creatine kinase (CK) activity on the 1st and 9th day of the training camp for the control (CON) and whole body cryotherapy exposure (WBC). The measurements between groups are compared by the unpaired version of *t*-test or the Mann–Whitney nonparametric test.

**Figure 3 ijerph-17-07855-f003:**
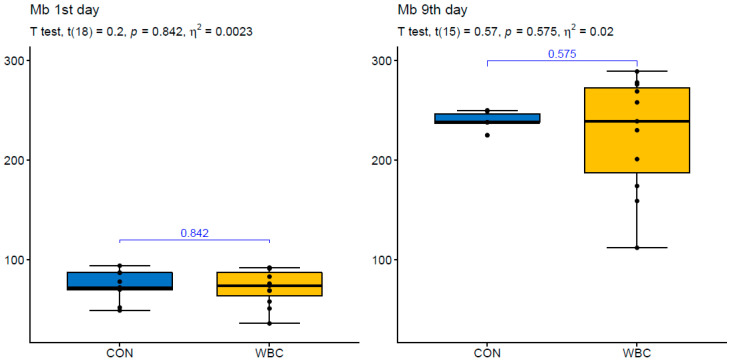
Visualization of statistical analysis of changes in myoglobin (Mb) concentration on the 1st and 9th day of the training camp for the control (CON) and whole body cryotherapy exposure (WBC). The measurements between groups are compared by the unpaired version of *t*-test or the Mann–Whitney nonparametric test.

**Table 1 ijerph-17-07855-t001:** Anthropometrics and body composition (mean ± SD).

	CON *n* = 9	WBC *n* = 11	CON vs. WBC
Age (years)	25.00 ± 2.83	24.27 ± 3.35	0.547
Height (cm)	172.00 ± 8.22	177.36 ± 7.15	0.192
Weight (kg)	78.76 ± 19.62	86.33 ± 20.65	0.373
BMI (kg/m^2^)	26.29 ± 3.98	27.17 ± 4.60	0.520
%FM	16.78 ± 4.95	11.59 ± 4.99	0.137
FM (kg)	13.80 ± 7.11	10.65 ± 7.50	0.580
FFM (kg)	64.96 ± 13.16	75.59 ± 14.60	0.098

Abbreviations: CON control group; WBC whole body cryotherapy group; BMI body mass index; FM fat mass; FFM fat-free mass. The measurements in groups are compared by the unpaired version of *t*-test or the Mann–Whitney nonparametric test.

**Table 2 ijerph-17-07855-t002:** The levels of oxi-inflammatory mediators.

	1st Day of Camp (Initial Level)			9th Day of Camp (After the Last WBC Session)			1st Day vs. 9th Day
	Mean ± SD	CON vs. WBC	*η* ^2^	Mean ± SD	CON vs. WBC	*η* ^2^	
H_2_O_2_ (µmol/L) CON WBC	25.00 ± 2.41 24.73 ± 5.22	0.888	0.001	28.11 ± 4.28 17.91 ± 4.41	*p* < 0.001	0.578	0.068 *p* < 0.05
NO (µmol/L) CON WBC	17.93 ± 1.36 18.77 ± 1.99	0.439	0.034	23.29 ± 3.35 13.69 ± 2.26	*p* < 0.001	0.744	*p* < 0.001 *p* < 0.001
IL-1β (pg/mL) CON WBC	1.46 ± 0.12 1.34 ± 0.26	0.208	0.087	1.87 ± 0.41 1.00 ± 0.15	*p* < 0.001	0.703	*p* < 0.05 *p* < 0.01
TNFα (pg/mL) CON WBC	2.54 ± 0.34 2.63 ± 0.39	0.990	0.000	3.08 ± 0.39 3.23 ± 0.55	0.584	0.017	*p* < 0.01 *p* < 0.001
hsCRP (mg/L) CON WBC	1.21 ± 0.09 1.19 ± 0.20	0.909	0.000	2.26 ± 0.40 1.62 ± 0.49	*p* < 0.001	0.562	*p* < 0.001 *p* < 0.01
HGF (pg/mL) CON WBC	734 ± 111 676 ± 68	0.396	0.041	1157 ± 97 1028 ± 108	*p* < 0.05	0.230	*p* < 0.001 *p* < 0.001
IGF-1 (ng/mL) CON WBC	160 ± 14 167 ± 31	0.554	0.020	192 ± 14 136 ± 28	*p* < 0.001	0.611	*p* < 0.001 *p* < 0.01
PDGF^BB^ (pg/mL) CON WBC	1596 ± 316 1735 ± 359	0.672	0.010	2116 ± 381 1455 ± 241	*p* < 0.001	0.664	*p* < 0.05 *p* < 0.05
VEGF (pg/mL) CON WBC	200 ± 63 170 ± 71	0.686	0.009	372 ± 84 221 ± 60	*p* < 0.001	0.550	*p* < 0.001 *p* < 0.05
BDNF (pg/mL) CON WBC	23,487 ± 2724 22,144 ± 4753	0.463	0.030	27,226 ± 2513 28,687 ± 5621	0.653	0.012	0.061 *p* < 0.01

Abbreviations: CON control group; WBC whole body cryotherapy group; H_2_O_2_ hydrogen peroxide; NO nitric oxide; IL-1β interleukin 1β; TNFα tumor necrosis factor α; hsCRP high sensitivity C-reactive protein; HGF hepatocyte growth factor; IGF-1 insulin-like growth factor 1; PDGFBB platelet-derived growth factor; VEGF vascular endothelial growth factor; BDNF brain-derived neurotrophic factor. *η*^2^ is a measure of effect size. Data in columns CON vs. WBC show the *p*-values of the *t*-test or the Mann–Whitney nonparametric test. The last column shows the *p*-values of the *t*-test or the Wilcoxon nonparametric test.

**Table 3 ijerph-17-07855-t003:** Relationships (Pearson’s correlation coefficient) between reactive oxygen and nitrogen species (RONS) and growth factors.

		HGF (pg/mL)	IGF-1 (ng/mL)	PDGF^BB^ (pg/mL)	BDNF (pg/mL)	VEGF (pg/mL)
CON	H_2_O_2_ (μmol/L)	0.521 *p* < 0.05	0.321 *p* = 0.194	0.151 *p* = 0.551	0.009 *p* = 0.997	0.323 *p* = 0.191
	NO (μmol/L)	0.620 *p* < 0.01	0.711 *p* < 0.01	0.597 *p* < 0.01	0.554 *p* < 0.05	0.419 *p* = 0.083
WBC	H_2_O_2_ (μmol/L)	−0.552 *p* < 0.01	0.473 *p* < 0.05	0.515 *p* < 0.05	−0.297 *p* = 0.179	−0.414 *p* = 0.055
	NO (μmol/L)	−0.686 *p* < 0.001	0.437 *p* < 0.05	0.449 *p* < 0.05	−0.502 *p* < 0.05	0.003 *p* = 0.989

Abbreviations: CON control group; WBC whole body cryotherapy group; H_2_O_2_ hydrogen peroxide; NO nitric oxide; HGF hepatocyte growth factor; IGF-1 insulin-like growth factor 1; PDGFBB platelet-derived growth factor; BDNF brain-derived neurotrophic factor; VEGF vascular endothelial growth factor.

**Table 4 ijerph-17-07855-t004:** Hematological and immunological variables.

	1st Day of Camp (Initial Level)			9th Day of Camp (After the Last WBC Session)			1st Day vs. 9th Day
	Mean ± SD	CON vs. WBC	*η* ^2^	Mean ± SD	CON vs. WBC	*η* ^2^	
HB (g/dL) CON WBC	15.39 ± 0.74 15.02 ± 1.12	0.838	0.005	14.64 ± 0.81 14.77 ± 1.07	0.250	0.073	0.093 0.253
RBC (mln/mm^3^) CON WBC	5.26 ± 0.34 5.18 ±0.29	0.564	0.019	5.01 ± 0.30 5.15 ± 0.25	0.246	0.075	0.157 0.567
HCT (%) CON WBC	44.90 ± 2.42 43.54 ± 2.78	0.484	0.028	43.21 ± 1.90 46.91 ± 2.98	*p* < 0.001	0.570	0.153 *p* < 0.001
MCV (fL) CON WBC	85.44 ± 2.60 84.06 ± 3.49	0.621	0.014	86.44 ± 2.35 46.91 ± 2.98	*p* < 0.001	0.617	0.032 *p* < 0.001
MCH (pg/RBC) CON WBC	29.28 ± 1.21 28.98 ± 1.48	0.918	0.000	30.12 ± 3.43 28.82 ± 1.54	0.665	0.011	0.479 0.445
MCHC (g/dL) CON WBC	34.24 ± 0.50 34.49 ± 0.78	0.425	0.036	33.84 ± 0.67 32.55 ± 1.99	*p* < 0.001	0.879	0.042 *p* < 0.001
PLT (10^3^/µL) CON WBC	249 ± 38 236 ± 48	0.200	0.090	246 ± 39 238 ± 53	0.746	0.006	0.620 0.727
LEU (10^3^/µL) CON WBC	6.41 ± 1.41 5.87 ± 0.89	0.282	0.064	5.96 ± 0.97 5.42 ± 0.93	0.226	0.081	0.451 0.420
LYM (10^3^/µL) CON WBC	2.17 ± 0.53 2.33 ± 0.30	0.404	0.039	2.24 ± 0.49 1.80 ± 2.26	*p* < 0.05	0.309	0.630 *p* < 0.001
NEU (10^3^/µL) CON WBC	2.11 ± 0.89 2.80 ± 0.72	0.093	0.266	2.05 ± 0.59 2.99 ± 0.90	*p* < 0.05	0.289	0.759 0.549
MON (10^3^/µL) CON WBC	0.41 ± 0.08 0.46 ± 0.06	0.141	0.087	0.41 ± 0.06 0.44 ± 0.12	0.488	0.026	0.972 0.591

Abbreviations: CON control group; WBC whole body cryotherapy group; HB hemoglobin; RBC red blood cells; HCT hematocrit; MCV mean cell volume; MCH mean corpuscular haemoglobin; MCHC mean corpuscular HB concentration; PLT platelets; LEU leucocytes; LYM lymphocytes; NEU neutrophils; MON monocytes. *η*^2^ is a measure of effect size. Data in columns CON vs. WBC show the *p*-values of the *t*-test or the Mann–Whitney nonparametric test. The last column shows the *p*-values of the *t*-test or the Wilcoxon nonparametric test.

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
