# Peer review of "Multiple Cryotherapy Attenuates Oxi-Inflammatory Response Following Skeletal Muscle Injury"

_ijerph, 2020, doi:10.3390/ijerph17217855_

Round 1

Reviewer 1 Report

The topic is interesting. Is is possible to detect which muscles are most interested in getting a benefit with multiple cryotherapy? Please look at these references: Anterior cervical corpectomy for cervical spondylotic myelopathy: Reconstruction with expandable cylindrical cage versus iliac crest autograft. A retrospective study. Clin Neurol Neurosurg. 2015 Dec;139:258-63. doi: 10.1016/j.clineuro.2015.10.023. Epub 2015 Oct 20. PMID: 26528885. AND Multiple cryotherapy applications attenuate oxidative stress following skeletal muscle injury. Redox Rep. 2017 Nov;22(6):323-329. doi: 10.1080/13510002.2016.1239880. Epub 2016 Oct 18. PMID: 27750503; PMCID: PMC6837703.

Author Response

Thank you for raising this question. It would have been interesting to explore. However, this time we did not expand our focus that far.

The proposed reference of Siqueira et al. Redox Rep 2017 has been included in Discussion [31].

Please forgive us but we could not include the reference to Perrini et al. Clin Neurol Neurosurg 2015 because that paper compares the clinical and radiographic outcomes between cervical reconstruction with expandable cylindrical cage and iliac crest autograft after one- or two-level anterior cervical corpectomy for spondylotic myelopathy. This departs from the subject matter which we investigated.

Reviewer 2 Report

In the current study, the authors examined the effect of whole-body cryotherapy on skeletal muscle damage, oxidative inflammatory mediators and hematological variables in male athletes participating a 14-day training session. The description was clear and results were stated precisely and straightforward. It is helpful that the authors compared the results in current study with those of pervious published literature. What is missing is the in-depth discussion about the mechanism(s) behind changes of the biomarkers, which may provide valuable insights in the improvement of this strategy in treating post-injury recovery. For example, previously published data in animal models may shed on light on the possible molecular mechanisms and be worthwhile to be discussed.  

Author Response

The following references concerning the effects of cold exposure on antioxidative markers in animal models have been added to Discussion: Siqueira et al. Redox Rep 2017 [31], Skrzep-Poloczek et al. Oxid Med Cell Longev 2017 [32] and Romuk et al. Biomed Res Int 2019 [33].

Reviewer 3 Report

This manuscript reads well. It is a limited study but was well executed. A few small changes are suggested.

Line 23. Superscript degrees symbol.

Line 64. Underlined reference 9. Possible leftover track changes.

Line 274. Typo? TNF (alpha).

It’s curious that the control participants were asked to sit whilst the WBC participants were instructed to walk slowly. I imagine the impact on results was minimal. Would the authors care to comment on this choice?

Please cite the software R correctly. This can be found with the function ‘citation()’ in the R software.

Author Response

Comments and Suggestions for Authors

This manuscript reads well. It is a limited study but was well executed. A few small changes are suggested.

Line 23. Superscript degrees symbol. Line 64. Underlined reference 9. Possible leftover track changes. Line 274. Typo? TNF (alpha).

All the above mentioned comments have been taken into account and changes have been made accordingly.

It’s curious that the control participants were asked to sit whilst the WBC participants were instructed to walk slowly. I imagine the impact on results was minimal. Would the authors care to comment on this choice?

Indeed our subjects from control group were asked to sit whilst the WBC participants were instructed to walk slowly because the cryotherapy was carried out in accordance with the protocol of Broatch et al. Sci Reports 2019 [17].  The method section has been completed with this information.

Please cite the software R correctly. This can be found with the function ‘citation()’ in the R software.

The method section concerning the software R has been completed with the information pointed out by the Reviewer: [20] R Core Team (2020). R: A language and environment for statistical computing. R Foundation for Statistical Computing, Vienna, Austria URL https://www.R-project.org/.

Reviewer 4 Report

In this study, Zembron-Lacny et al. performed a study to examine the impact of whole-body cryotherapy on circulating mediators of skeletal muscle regeneration , including serum CK and myoglobin, oxi-inflammatory factors H2O2 and NO, as well as various haematological and immunological parameters. The study was properly designed and conducted, and results were clearly interpreted. This study to some degree clarified the role of WBC in atheletes following skeletal muscle injury, and could impact the practice in sports medicine.

I only have a few minor comments:

  1. Line 23 - "3-min exposure to 120oC", o should be superscript.
  2. Line 86 - please clarify if "National Centre for Sports Medicine" refers to National Centre for Sports Medicine of Poland.
  3. Lines 88-89 - please specify if "National Olympic Sports Centre" refers to National Olympic Sports Centre of Poland.
  4. Line 124 - please revise "between 7.00 and 8.00 a.m." to "between 7:00 and 8:00 a.m."
  5. Lines 170-173 - "The markers of muscle damage in the study athletes tended to reach lower values after 7-day WBC exposure compared to the control group thus the values did not differ considerably. This indicates that repeated cryotherapy did not affect the extent of tissue damage in athletes." The first sentence is confusing, as there's a significant decrease of serum CK on the 9th day (right panel of Figure 2). I also do not understand how does this indicate "repeated cryotherapy did not affect the extent of tissue damage". Please explain.
  6. Lines 281-282 - please revise "heat-stress proteins" to "heat shock proteins".

Author Response

1. Line 23 - "3-min exposure to 120oC", o should be superscript; Line 86 - please clarify if "National Centre for Sports Medicine" refers to National Centre for Sports Medicine of Poland; Lines 88-89 - please specify if "National Olympic Sports Centre" refers to National Olympic Sports Centre of Poland; Line 124 - please revise "between 7.00 and 8.00 a.m." to "between 7:00 and 8:00 a.m."; Lines 281-282 - please revise "heat-stress proteins" to "heat shock proteins".

Following the Reviewer’s suggestion, all the corrections have been made.

2. Lines 170-173 - "The markers of muscle damage in the study athletes tended to reach lower values after 7-day WBC exposure compared to the control group thus the values did not differ considerably. This indicates that repeated cryotherapy did not affect the extent of tissue damage in athletes. " The first sentence is confusing, as there's a significant decrease of serum CK on the 9th day (right panel of Figure 2). I also do not understand how does this indicate "repeated cryotherapy did not affect the extent of tissue damage". Please explain.

Thank you for this suggestion. Both sentences have been rewritten and the revised text reads as follows: CK activity reached lower values while Mb concentration did not differ after 7-day WBC exposure compared to the control group. This indicates that repeated cryotherapy did not have a considerable impact on the extent of tissue damage in athletes.”

Round 2

Reviewer 1 Report

Well done.